# Effect of Vanillin on the Anaesthesia of Crucian Carp: Effects on Physiological and Biochemical Indices, Pathology, and Volatile Aroma Components

**DOI:** 10.3390/foods12081614

**Published:** 2023-04-11

**Authors:** Lexia Jiang, Jiaming Tang, Baosheng Huang, Changfeng Zhang, Peihong Jiang, Dongjie Chen

**Affiliations:** 1Key Laboratory of Agricultural Products Storage, Transportation and Preservation Technology of Shandong Province, Shandong Institute of Commerce and Technology, Jinan 250103, China; 13977695205@163.com (L.J.); tang19980503@163.com (J.T.); zcf202@163.com (C.Z.); jiangpeihongjph@163.com (P.J.); dongjie613@163.com (D.C.); 2College of Food Science and Technology, Guangdong Ocean University, Zhanjiang 524088, China; 3National Agricultural Products Modern Logistics Engineering Technology Research Center, Jinan 250103, China; 4Shandong Guonong Logistics Technology Co., Ltd., Jinan 250103, China

**Keywords:** vanillin, crucian carp, anesthetic effect, physiological and biochemical indices, pathology, volatile components

## Abstract

The anaesthetic effect of vanillin on crucian carp was investigated using different concentrations of vanillin, with a nonvanillin control. The effective concentration range of vanillin anaesthesia was determined from the behavioural characteristics of crucian carp during the anaesthesia onset and recovery phases. Physiological and biochemical indices, and the electronic nose response to the fish muscle, were measured over the range of effectiveanaestheticc concentrations. An increased concentration of vanillin shortened the time taken to achieve deep anaesthesia but increased the recovery time. The levels of white blood cells, red blood cells, haemoglobinn, platelets, alanine aminotransferase, alkaline phosphatase, lactate dehydrogenase, phosphorus, potassium, magnesium, total protein, and serum albumin were lower than the control in the vanillin treatment group. Triglycerides and total cholesterol were not significantly affected. Histology showed no effect of vanillin on the liver, except at 1.00 g/L vanillin. Vanillin resulted in a nondose-responsive effect on the gill tissue, increasing the width and spacing of the gill lamellae. E-Nose analysis of the carp-muscle flavour volatiles was able to distinguish between different vanillin treatment concentrations. GC-IMS identified 40 flavour compounds, including 8 aldehydes, 11 alcohols, 10 ketones, 2 esters, and 1 furan. Vanillin had aanaestheticic effect on crucian carp and these findings provide a theoretical basis for improving the transport and experimental manipulation of crucian carp.

## 1. Introduction

Freshwater fish are popular with consumers for their high-protein and low-fat contents, and there is increasing consumer demand for high-quality fish. During the production and transportation of fish, trauma and stress often occur, affecting their quality and survival rate. Fish anaesthetics are widely used, as they have a sedative effect during transport, and effectively reduce fish mortality. Commonly used fish anaesthetics in China are ethyl m-amino benzoate (MS-222), eugenol, and 2-phenoxyethanol, however, all three anaesthetics have undesirable side effects. MS-222 solution is weakly acidic, resulting in the plasma-cortisol content increasing after the fish recover from deep anaesthesia; exposure to direct sunlight may produce toxicity to fish when using MS-222 [1]. Eugenol is volatile and its effect diminishes during anaesthesia [2,3]; 2-Phenoxyethanol is hazardous to fishery workers, causing neurological syndromes after prolonged exposure, has a severe, long-lasting residual effect, and remains active for three days after the fish recover [4]. Therefore, there is a need for research into safer anaesthetics for fishery use, such as those of a plant origin, which have antimicrobial and antioxidant activities, and carry a low risk to human health [5,6]. Plant extracts, such as alfalfa essential oil [7], thyme essential oil [8], and menthol [9] all have anaesthetic effects on fish. 

Vanillin is the common name for 3-methoxy-4-hydroxybenzaldehyde, which is extracted from the seeds of orchids of the genus *Vanilla* [10]. Annual consumption of vanillin in China is 2000–2500 t, with consumption increasing every year [11]. Vanillin has biological activities, such as antibacterial, anti-inflammatory, and sedative. Vanillin improved the resistance of papaya to invasion by pathogenic bacteria, thereby improving fruit quality and improved resistance to papaya postharvest rot; vanillin can be used as a safe, nontoxic inducer and preservative to inhibit the growth of rot pathogens [12]. Vanillin encapsulated in nanofiber membranes extended the shelf life of turbot fillets at 4 °C [13]. Inhalation of vanillin by mice had no effect on activity and cognitive function, though inhalation of essential oils containing vanillin produced effective sedation [14]. Vanillin alleviated rotational behavioural symptoms, reduced dopamine neuron damage, inhibited microglial activation in a rat model, and had a therapeutic effect in an animal model of lipopolysaccharide-induced Parkinson’s disease [15]. The antimicrobial activity of vanillin reduced the heat resistance of *E. coli* [16]. Vanillin has human health-promoting and medicinal properties and is also an important raw material for the synthesis of many drugs. Vanillin has been reported to reduce infarct volume after hypoxic-ischaemic brain injury, by reducing blood–brain barrier damage and oxidative damage [17]. Vanillin has neuroprotective effects by regulating the expression of inflammatory cytokines in mice with transient middle cerebral artery occlusion, demonstrating that vanillin can inhibit the TLR4/NF-κB signalling pathway by an anti-inflammatory mechanism [18]. Vanillin improved survival and reduced oxidative stress in rats suffering from sepsis [19]. In this study, the anaesthetic effect of vanillin was tested on crucian carp and the resulting changes in physiological and biochemical parameters, pathology and volatile aroma compounds were investigated.

## 2. Materials and Methods

### 2.1. Materials and Reagents

Live crucian carp (*Carassius auratus*) from the Jinan seafood market (Jinan, China); fish were selected for good health and active movement. The fish were maintained in a temperature-controlled circulating water filtration system (dissolved oxygen ≥ 7 mg/L, pH 7–8), at the National Agricultural Products Modern Logistics Engineering Technology Research Center. Vanillin (purity ≥ 99%), was from Sinopharm Chemical Reagents (Shanghai, China). Potassium (K^+^), sodium (Na^+^), glucose (GLU), total protein (TP), albumin (ALB), total cholesterol (CHO), and triglyceride (TG) assay kits were all from Shandong Boke Biological Industries (Jinan, China). Xylene and neutral resin were from National Group Chemical Reagents (Shanghai, China). Hematoxylin and eosin (H&E) staining solution, differentiation solution, and blue return solution were from Wuhan Xavier Biotechnology (Wuhan, China). Ethylene glycol ethyl ether acetate was from Shanghai Macklin Biochemical Technology (Shanghai, China).

### 2.2. Experimental Methods

#### 2.2.1. Determination of Effective Concentration of Vanillin for Anaesthesia of Crucian Carp

The effective concentration for this determination was defined as the onset of anaesthesia within 3 min and awakening within 5 min [20]. Six treatment concentrations (0.25, 0.50, 0.75, 1.00, 1.25, and 1.50 g/L) were used and 60 carp were randomly selected in total for the treatment and control groups. When the fish entered the anaesthesia phase (A_3_), the corresponding time was recorded as anaesthesia onset, then the fish were immediately transferred to fresh water for recovery and the time was recorded as the onset of recovery phase 3 (R_3_).

#### 2.2.2. Determination of Behavioural Characteristics during the Recovery Phases of Anaesthesia, at the Optimal Vanillin Concentration

When the anaesthetic concentration is low, the fish cannot enter anaesthesia quickly; when the anaesthetic concentration is high, the fish enter anaesthesia too quickly and the behavioural characteristics of the anaesthesia are difficult to distinguish. To observe stable and distinct anaesthetic staging characteristics, an appropriate anaesthetic concentration is required. Based on the results from Section 2.2.1, the behavioural characteristics of the carp during anaesthesia were observed in a 1.00 g/L vanillin solution. The anaesthesia (A_0_–A_4_) and recovery (R_1_–R_4_) periods were classified according to the criteria of Mirghaed et al. [21] and correlated with the behavioural characteristics of the different periods.

#### 2.2.3. Effect of Vanillin Concentration on Blood and Serum Parameters of Crucian Carp

Three crucian carp were randomly placed in treatment (different vanillin concentrations) and control tanks, then each fish was tested for anaesthesia (each fish was tested only once). The fish were kept in a state of complete anaesthesia for 10 min, and then blood samples (5 mL) were drawn with a syringe from the tail vein. Part of the sample was mixed with an anticoagulant and used for the determination of the physiological parameters, red blood cells (RBC), white blood cells (WBC), haemoglobin (HGB), and platelets (PLT); the other part was centrifuged at 4 °C for 20 min and 4000 rpm. Then, the supernatant was stored at −80 °C until needed for the determination of serum alanine aminotransferase (ALT), aspartate aminotransferase (AST), alkaline phosphatase (ALP), lactate dehydrogenase (LDH), inorganic phosphate (P_i_), magnesium (Mg^2+^), potassium (K^+^), sodium (Na^+^), glucose (GLU), total protein (TP), albumin (ALB), cholesterol (CHO), and triglycerides (TG). 

#### 2.2.4. H&E Staining and Histological Examination of Liver and Gill Tissues

The carp were put under complete anaesthesia with different concentrations of vanillin, maintained for 10 min and then gently removed and placed on a dissecting table to remove the liver and gill parts for H&E staining. All samples were cut into 5–6 µm thick slices. Then H&E was stained for microscopic observation. Dehydration and transparency were performed in ethanol and xylene, and then cover slipped. The stained sections were observed and photographed using a light microscope (Nikon, DS-Fi2) and spliced into a complete image using ImageJ software.

#### 2.2.5. E-nose Analysis

The carp were put under complete anaesthesia with different concentrations of vanillin, maintained for 10 min, and then gently removed and placed on a dissecting table to remove the muscle for the electronic nose test. The E-Nose (FOX4000, Alpha MOS, Toulouse, France) sampling was performed as described previously [22], with some modifications. Muscle tissue (5.0 g) was homogenized, then sealed into 10 mL sample vials and analyzed each sample 4 times. The samples were analyzed as follows: carrier gas velocity, 150 mL/min; headspace generation temperature, 40 °C; injection volume, 2 mL; injection speed, 2 mL/s; headspace generation time, 600 s; data acquisition time, 120 s; delay time, 400 s. Radar charts and linear discriminant analysis (LDA) were used to analyse the data and remove outliers. The response characteristics of each E-Nose sensor are shown in Table 1.

#### 2.2.6. Gas Chromatography-Ion Mobility Spectrometry (GC-IMS) Analysis of Flavour Volatile Compounds

The carp were put under complete anaesthesia with different concentrations of vanillin, maintained for 10 min and then gently removed and placed on a dissecting table to extract muscle parts for GC-IMS studies. The analysis was performed with a FlavourSpec flavour analyser (G.A.S., Dortmund, Germany). The control software has built-in NIST and IMS databases, which were used for analyte identification. A 4.0 g sample was placed in a 20.0 mL headspace vial and incubated for 15 min at 60 °C. Headspace sampling conditions were as follows: headspace temperature, 60 °C; incubation time, 15 min; heating mode, oscillation; headspace injection needle temperature, 85 °C; injection volume, 500 μL; nonsplit mode; carrier gas, N_2_ (purity ≥ 99.999%); and wash time 0.50 min. Chromatography conditions were as follows: column temperature, 60 °C; run time, 15 min; carrier gas, N_2_ (≥99.999%); flow rate: initially 5.0 mL/min, held for 10 min, and then linearly increased to 150 mL/min over 5 min. Detection conditions were as follows: drift tube length, 5 cm linear voltage in the tube, 400 V/cm; drift tube temperature, 40 °C; drift gas, N_2_, (≥99.999%); flow rate, 150 mL/min; and IMS detector temperature, 45 °C. 

#### 2.2.7. Statistical Analysis

SPSS 26.0 (SPSS Inc., Chicago, IL, USA) was used for data analysis. Data are expressed as the mean ± standard deviation, according to an analysis of variance (ANOVA) and Duncan’s multiple-range test (*p* < 0.05). The figures were plotted using OriginPro 2021 (OriginLab, Northampton, MA, USA). GC-IMS data were collected and analyzed from different perspectives using the Laboratory Analytical Viewer (LAV) software and three plug-ins (Reporter, Gallery Plot, Dynamic PCA) as well as the GC-IMS Library Search qualitative software.

## 3. Results

### 3.1. Determination of the Effective Vanillin Anaesthesia Concentration Range for Crucian Carp 

As the concentration of vanillin increased, the time taken to achieve deep anaesthesia (stage A5; anaesthesia time) decreased and the time for complete recovery increased (Table 2, Figure 1); the anaesthesia time with 0.25 g/L vanillin was markedly longer than at other concentrations. The average anaesthesia time at 1.50 g/L was 50% of that at 0.50 g/L, whereas the average recovery time was 281% of that at 0.50 g/L. The recovery rate of carp was 100% at all six vanillin concentrations, indicating that, within the concentration range of 0.50–1.50 g/L, the vanillin bath was safe for the fish. As the vanillin concentration increased, the anaesthesia time decreased, in a dose-responsive, though nonlinear manner (Table 2). Taking market application as the starting point, the short average anaesthesia time is the focus of market demand, and it can be seen that the anaesthesia efficiency is higher when the test concentration is 0.50 g/L~1.50 g/L.

#### Behavioural Characteristics of Crucian Carp during the Five Stages of Anaesthesia Onset and the Four Stages of Recovery

The timing and behavioural characteristics at each stage of anaesthesia (A_0_–A_4_) and recovery (R_1_–R_4_) of crucian carp were determined using 1.00 g/L vanillin (Table 3). When a fish stopped swimming, completely lost the ability to respond, and the operculum opened weakly and slowly, it was defined as being at the A_3_ stage of the anaesthesia period, and when the fish body was balanced and responsive, and the operculum opened normally, it was defined as being at the R_3_ stage of recovery.

### 3.2. Effect of Vanillin Concentration on Blood and Serum Parameters of Crucian Carp

#### 3.2.1. Effect of Vanillin Concentration on the Blood Composition Index of Crucian Carp

The control WBC count was 809.76 ± 21.51 × 10^9^/L; the vanillin treatment WBC count initially markedly decreased, then slightly increased, with the increased vanillin concentration (Figure 2A). The control RBC count was 0.17 ± 0.10 × 10^12^/L and markedly decreased at all vanillin treatment concentrations, with a minimum of 0.75 g/L (Figure 2B). The control HGB content was 95.11 ± 4.59 g/L and moderately decreased, then slightly increased, with increased vanillin treatment concentration (Figure 2C). The control blood PLT content was 27.22 ± 4.63 × 10^9^/L and markedly decreased at all vanillin treatment concentrations; the PLT content was not significantly different between the five vanillin concentrations (Figure 2D).

#### 3.2.2. Effect of Vanillin Concentration on Blood Serum Enzymes in Crucian Carp

The control serum ALT concentration was 97.73 ± 58.18 U/L; the vanillin treatment ALT concentration initially markedly decreased, then slightly, though not significantly, increased, with increased vanillin concentration, except for a marked increase at 1.50 g/L vanillin, to 111.77 ± 21.97 U/L (Figure 3A). The control serum AST concentration was 440.4 ± 177.39 U/L; the vanillin treatment AST concentration markedly increased, though remained relatively constant, at all vanillin treatment concentrations (Figure 3B). The control serum ALP concentration was 26.67 ± 12.66 U/L; the vanillin treatment ALT concentration initially markedly decreased, then gradually increased, with the increased vanillin concentration (Figure 3C). The control serum LDH concentration was 2560.02 ± 7.23 U/L and moderately decreased at all vanillin treatment concentrations, with a minimum of 1.00 g/L (Figure 3D).

#### 3.2.3. Effect of Vanillin Concentration on the Blood Serum Ion Concentrations of Crucian Carp

The PO_4_^3−^ concentration gradually decreased from the control value with the increased vanillin concentration, with a minimum of 1.25 g/L, and a significant increase at 1.50 g/L (Figure 4A). The K^+^ concentration markedly decreased from the control value with increased vanillin concentration, with a minimum of 1.00 g/L, and a significant increase at 1.50 g/L (Figure 4B). The Mg^2+^ concentration gradually decreased from the control value with increased vanillin concentration, with a minimum of 1.25 g/L, and a slight increase of 1.50 g/L (Figure 4C). The Na^+^ concentration significantly increased from the control value at all vanillin treatment concentrations, with a gradual increase with increased vanillin concentration to a maximum of 1.25 g/L, and a small decrease at 1.50 g/L (Figure 4D).

#### 3.2.4. Effect of Vanillin Concentration on the Blood Serum Concentrations of Organic Components in Crucian Carp

The control TP concentration of 76.33 ± 6.79 g/L decreased gradually with increasing vanillin treatment concentration to a minimum of 1.00 g/L vanillin, then gradually increased; the variation in ALB content was very similar to that of the TP content (Figure 5A). The CHO concentration of the vanillin treatments was slightly higher than the control (Figure 5B). The GLU concentration varied widely with increased vanillin concentration, increasing markedly at 0.50 g/L with a further increase at 0.75 g/L, followed by a marked decrease to a minimum at 1.25 g/L, and then a small increase at 1.50 g/L (Figure 5C). The control TG concentration slightly increased at 0.50 g/L vanillin, then slightly decreased with increasing vanillin concentration (Figure 5D).

### 3.3. Histopathological Examination

Liver tissue sections (Figure 6) showed that the hepatocytes in the control and treatment groups were evenly arranged, intact and clear, except at 1 g/L vanillin, and vacuolation in the treatment groups was more marked than in the control. At 1.50 g/L vanillin, the accumulation of hepatocyte nuclei was greater than in the control, though their integrity was better.

Gill tissue sections (Figure 7) showed that the control gill lamellae (GL) were flattened, vesicular, elongated, and arranged in a comb-like pattern, they consisted of a single layer of epithelial cells, capillaries, and columnar support cells. The epithelial cells were flattened with spindle-shaped nuclei, and the columnar support cells had large, round nuclei and were attached to the basement membrane. The filaments and lamellae contained small numbers of chlorine-secreting cells. Compared to the control, the filaments and lamellae were swollen at 1.50 g/L vanillin; there was a proliferation of flattened and columnar cells in the vascular walls of the filaments, an increase in chlorine-secreting cells, curling and fusion of the lamellae, and detachment of the supragillar cells. Microstructural measurements showed a slight swelling of the chlorine-secreting cells at all treatment concentrations, which was not significantly different from the control, except at 1.00 g/L, at which the width and spacing of the gill lamellae increased significantly.

### 3.4. Effect of Vanillin Concentration on the Volatile Flavour Compound Profile of Carp Muscle, Determined by E-Nose Analysis

#### 3.4.1. Radar Fingerprinting of the E-Nose Sensor Responses to Carp-Muscle Flavour Volatiles

The muscle aroma of the anaesthetized crucian carp was analyzed at different vanillin treatment concentrations. The 18 sensors of the FOX4000 E-Nose are sensitive to different chemical classes of volatile compounds and the ratio of relative conductivity G/G_0_ is proportional to the concentration of the corresponding compound class. When G/G_0_ > 1, the volatile concentration is detectable and when G/G_0_ ≤ 1 the volatile concentration is below the detection limit [23]. There was a wide variation in sensor response (Figure 8), with six sensors giving no response and those of T30/1, P10/1, PA/2, and P30/1 being relatively strong. However, the differences between the control and the six vanillin treatment concentrations were very small. It appears that vanillin treatment has a negligible effect on fish flavour, which is highly desirable from a commercial viewpoint.

#### 3.4.2. Linear Discriminant Analysis (LDA) of E-Nose Data

LDA is a statistical method that uses samples of known categories to establish a discriminant model and discriminate samples of unknown categories [24]. The E-Nose data were subjected to an LDA dimension reduction analysis (Figure 9). The contributions of LDA1 and LDA2 were 77.29% and 19.69%, respectively, and the combined contribution was 96.98%. LDA can distinguish the treatment and control samples, though there was no dose response to the vanillin treatment. 

### 3.5. GC-IMS Analysis of Aroma Volatiles from Crucian Carp Muscle at Different Vanillin Treatment Concentrations 

The GC-IMS 3D spectra of carp-muscle flavour volatiles resulting from different vanillin treatment concentrations (Figure 10) are not easily distinguished by the eye.

The 2D top-view GC-IMS plots (Figure 11A,B) allow visual comparison of the volatile flavour profiles of carp muscle resulting from different vanillin treatment concentrations and significant differences between samples are visible. 

The GC retention time and IMS migration time of the volatile flavour compounds from carp muscle after treatment, with different concentrations of vanillin, were compared with the GC-IMS database on crucian carp muscle volatiles. The library search identified 40 flavour compounds (Figure 12, Table 4), including eight aldehydes, 11 alcohols, 10 ketones, two esters, and one furan.

To analyze differences in aroma profiles resulting from different vanillin treatment concentrations, a flavour profile fingerprint was generated from the triplicate analyses of the control and vanillin treatment samples (Figure 13). Some compounds, such as 3-hydroxy-2-butanone, 2,3-butanedione, 2,3-pentanedione, 2,3-butanediol, ethanol, and 2-methylbutanol were present in similar amounts in all samples. Some other compounds varied in content greatly between different samples, such as hexanal, E-2-hexenal, methyl acetate, 2-heptanone, 3-octanone, 4-methyl-3-penten-2-one, 2-pentylfuran, and 5-methylfuran, which were higher in A or D and lower in C, E, and F. The difference in muscle volatile flavour substances between group A and group D was not significant, indicating that vanillin with a mass concentration of 1 g/L had little effect on the muscle flavour of the crucian carp.

## 4. Discussion and Conclusions

Vanillin had an anaesthetic effect on crucian carp when added to the water in which the fish were present. With an increasing vanillin concentration, the time taken to achieve deep anaesthesia decreased in an exponential, dose-responsive manner and the recovery time of the fish, when removed to vanillin-free water, increased. All the test fish recovered completely and there was no evidence of adverse effects. This is consistent with reports on the anaesthetic effect of eugenol on juvenile yellow-spotted basketfish (*Siganus oramin*) [25], magnolia essential oil on spotted seabass (*Lateolabrax maculatus*) [26], and carbon dioxide on juvenile oval pompano trevally (*Trachinotus ovatus*) [27]. 

Blood physiological indices in fish are good indicators of their metabolic state, nutritional status, and health. White blood cells (WBC) are an essential part of the immune system [28]; red blood cells (RBC) deliver oxygen to body tissues [29]; platelets (PLT) are mainly involved in blood clotting, though they also have immunomodulatory effects; and haemoglobin HGB is the oxygen-binding protein in RBCs [30]. Liang et al. [31] analysed the anaesthetic effect of clove oil on the blood indexes of tilapia; the content of WBC, RBC, HGB, and PLT increased after anaesthesia. In this study, vanillin-induced anaesthesia moderately reduced the contents of WBC and HGB, though more than halved the contents of RBC and PLT.

Blood serum indices in fish are also good indicators of their metabolic state, nutritional status, health, and of cell membrane integrity [32]. Alanine and aspartate aminotransferases (ALT and AST) are important mitochondrial enzymes and are abundant in hepatocytes [33]. Under normal conditions, only small amounts of aminotransferases are released into the blood from hepatocytes and, therefore, serum aminotransferase activity is low. Elevated serum levels of ALT and AST are indicators of liver damage. Alkaline phosphatase (ALP) is involved in metabolism and the immune system [34]. Lactate dehydrogenase (LDH) is abundant in cardiac muscle cells and increased activity is an indicator of damage to the cardiomyocytes [35]. In this study, the ALT activity of carp anaesthetized with vanillin was reduced, whereas the AST activity markedly increased, indicating that there may have been some liver damage in the fish, which is consistent with the effect of anaesthesia with *Buddleja lindleyana* on serum AST in crucian carp [36]. The LDH activity was moderately reduced, suggesting that anaesthesia reduced myocardium damage, consistent with the response of MS-222 anaesthesia on bream (*Parabramis pekinensis*) [37]. The ALP activity of crucian carp anaesthetized by the various concentrations of vanillin solution was lower than that of the control group, indicating that the vanillin aqueous solution contained components that inhibited ALP activity, indicating that vanillin had less effect on ALP in crucian carp.

Inorganic ions such as PO_4_^3−^, Mg^2+^, K^+^, and Na^+^ in serum are important for maintaining osmolality, acid-base balance, and overall homeostasis in fish, and their content changes after environmental stress [38]. Elevated serum PO_4_^3−^ and Mg^2+^ levels in fish can indicate kidney damage [39]. In this study, the levels of PO_4_^3−^ and Mg^2+^ in anaesthetized crucian carp were reduced, indicating that vanillin did not cause kidney damage. The increase in serum Na^+^ and the decrease in serum K^+^ suggest increased permeability of the gill epithelium, which allows leakage of Na^+^ and ingress of K^+^. In this study, with the increase in the vanillin mass concentration, the Na^+^ content of each concentration of the anaesthesia group increased compared with the control group, and the K^+^ content decreased compared with the control group, and it was speculated that the higher concentrations of vanillin caused increased gill tissue cell deformation. This is consistent with the anaesthesia of tilapia with MS-222, which decreased serum K^+^ in a dose-responsive manner, whereas the Na^+^ concentration remained relatively stable [40].

Glucose in the bloodstream supplies energy for various vital activities in fish [41] and its concentration is regulated by insulin and epinephrine and fluctuates in response to environmental factors [42]. In this study, the treatment group’s blood glucose levels were higher than the control in different proportions, possibly due to less dissolved oxygen in the water, which lowered the metabolic rate and/or because of the reduced physical activity of the anaesthetized fish. The increases in GLU concentration after anaesthesia may result from a need for increased energy associated with the stress response, which is met by increased glycogenolysis. At higher vanillin concentrations, there was an irregular decrease in blood glucose, consistent with more rapid anaesthesia, which is consistent with a previous report on electroanesthetized pearl gentian grouper [43]. In contrast, blood glucose concentrations decreased after CO_2_ anaesthesia of the grouper [44]. Serum albumin (ALB) is the most abundant serum protein and acts as a transporter for insoluble fatty acids [45]. The ALB concentration was slightly decreased by vanillin treatment. Cholesterol (CHO) is an important component of the cell membrane and is transported in the bloodstream by lipoproteins in fish [46]. Triglycerides, cholesterol, and total protein levels are affected by protein catabolism and hepatic glycogenolysis. In this study, except for triglycerides, the TP and ALB contents of the crucian carp after anaesthesia were lower than those in the control group, indicating that the protein breakdown and liver glycogen decomposition of crucian carp decreased after anaesthesia.

Fishery anaesthetics are a class of substances that inhibit the sensory centres of the fish brain to varying degrees, causing the fish to lose the ability for reflex action. The mechanism of action is to first inhibit the brain cortex (tactile loss phase), then act on the basal ganglia and cerebellum (excitation phase), and finally on the spinal cord (anaesthesia phase) [47]. An appropriate dose of anaesthetic reduces oxygen consumption and ammonia excretion, inhibits excessive stress in the fish, and effectively reduces injury, or mortality during handling. The liver is an important glandular and digestive-metabolic organ in fish and is involved in bile secretion, metabolism, detoxification, and defence [48]. The gills are the main respiratory organ of fish and excrete metabolic wastes, such as ammonia and nitrogen [49]. Anaesthesia, with 40 mg/L MS-222 for 24 h, may cause some damage to the liver tissue of the *Larimichthys crocea* [50]; eugenol anaesthesia of the carp (*Cyprinus carpio*) did not result in detectable liver or kidney damage [51]; deep anaesthesia of rainbow trout with *Coriandrum sativum* essential oil did not result in gill or liver damage [52]. In this study, vanillin caused no detectable pathological effects on the liver. All concentrations of vanillin, except for 1.00 g/L, had some effect on the gill tissue, increasing the width and spacing of the gill lamellae. Further research will be needed to determine whether this is harmful to the fish and the mechanism of any harm.

The electronic nose (E-Nose) is an array of gas sensors that mimics the human olfactory system and characterizes the aroma of a sample, which has low cost, ease of operation, and high accuracy [53]. E-Nose analysis of carp muscle after vanillin treatment indicated that the most abundant flavour volatiles are alcohols, amines, and hydrocarbons. Linear discriminant analysis separated the control and the different vanillin concentration treatments, though there was no clear dose response in the separation, indicating that, although the E-Nose could discriminate the control and treatment samples, there was little difference in the flavour profiles between the different vanillin treatment concentrations. The response of different concentrations of vanillin to the muscle of crucian carp. Gas-phase ion mobility spectrometry (GC-IMS) combines gas chromatography and ion mobility spectrometry [54] and has the advantages of the high resolution of ion mobility spectrometry and the high sensitivity of gas chromatography, resulting in richer chemical information than GC-mass spectroscopy [55]. In this study, the GC-IMS was used to analyze differences in carp-muscle flavour profile after anaesthesia with different vanillin concentrations. A total of 40 flavour compounds were identified, including eight aldehydes, 11 alcohols, 10 ketones, two esters, and one furan. Of these, 3-hydroxy-2-butanone, 2,3-butanedione, 2,3-pentanedione, 2,3-butanediol, ethanol, and 2-methylbutanol were present in all samples with similar signal intensities, suggesting that these compounds are the main contributors to the overall flavour of crucian carp muscle tissue after anaesthesia. Moreover, the difference between the control group and the vanillin concentration of 1 g/L on muscle volatile flavour substances was not significant, indicating that vanillin with a mass concentration of 1 g/L had little effect on the muscle flavour of crucian carp.

In conclusion, the effective concentration range of vanillin to anaesthetize crucian carp is 0.50–1.50 g/L, which resulted in relatively small changes to blood biochemistry, and no detectable liver damage, though minor, nonlethal damage to the gills. Future research on the anaesthetic effect of vanillin should include an examination of molecular toxicology and protein metabolism. GC-IMS proved very useful for the identification of carp-muscle flavour volatiles, however, the GC-IMS database is not yet complete for fish muscle flavour compounds, so more information on these compounds will be needed.

## Figures and Tables

**Figure 1 foods-12-01614-f001:**
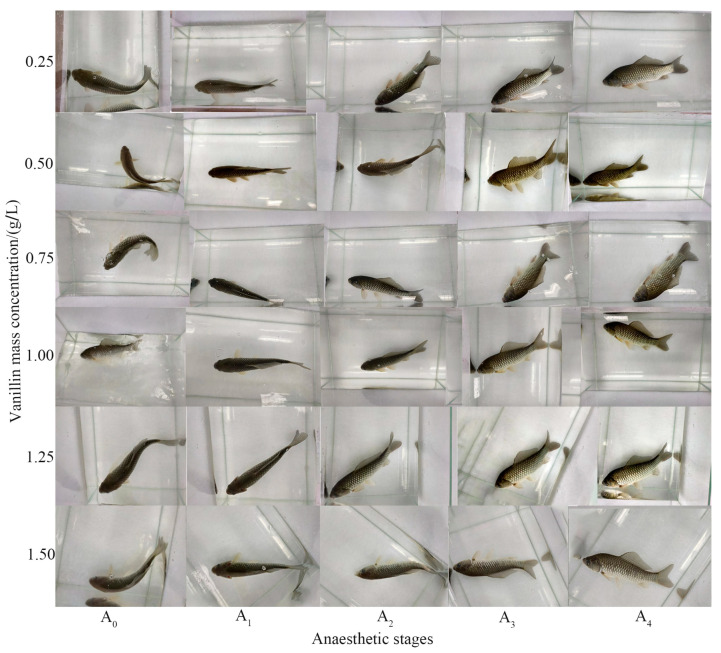
Appearance of crucian carp at the five stages of anaesthesia.

**Figure 2 foods-12-01614-f002:**
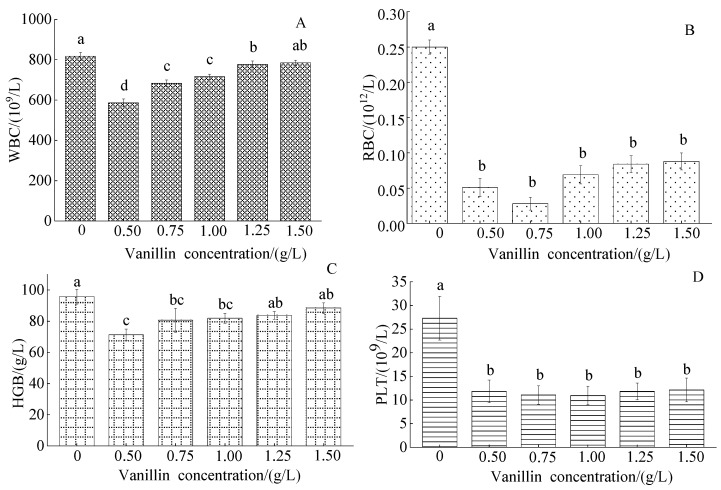
Effect of vanillin concentration on the blood composition of crucian carp: (**A**) white blood cells (WBC); (**B**) red blood cells (RBC); (**C**) haemoglobin (HGB); and (**D**) platelets (PLT). Different letters above the columns indicate a significant difference (*p* < 0.05); the same letter indicates no significant difference (*p* > 0.05).

**Figure 3 foods-12-01614-f003:**
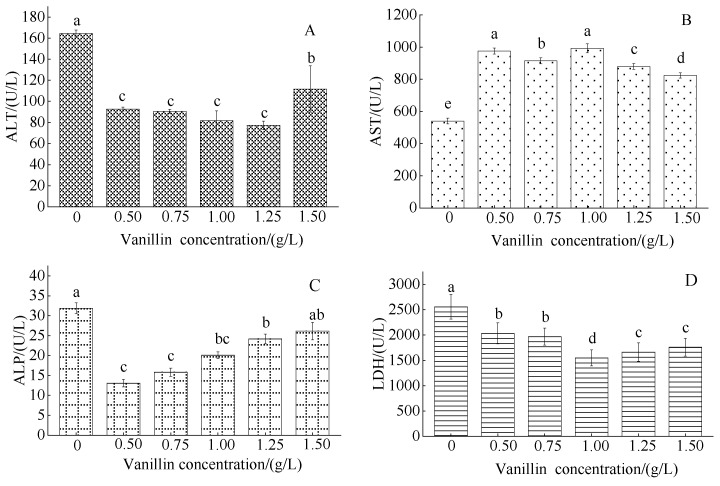
Effect of vanillin concentration on serum enzyme concentrations in crucian carp: (**A**) alanine aminotransferase (ALT); (**B**) aspartate aminotransferase (AST); (**C**) alkaline phosphatase (ALP); and (**D**) lactate dehydrogenase (LDH). Different letters above the columns indicate a significant difference (*p* < 0.05); the same letter indicates no significant difference (*p* > 0.05).

**Figure 4 foods-12-01614-f004:**
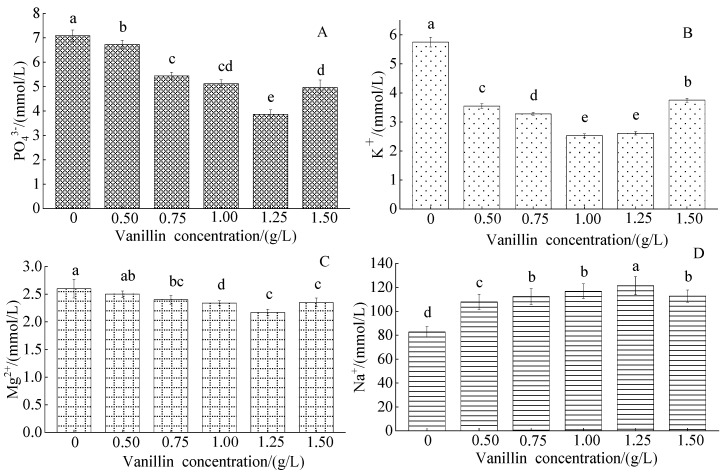
Effect of vanillin concentration on the blood serum ion content of crucian carp: (**A**) phosphate (PO_4_^3−^); (**B**) potassium (K^−^); (**C**) magnesium (Mg^2+^); and (**D**) sodium (Na^+^). Different letters above the columns indicate a significant difference (*p* < 0.05); the same letter indicates no significant difference (*p* > 0.05).

**Figure 5 foods-12-01614-f005:**
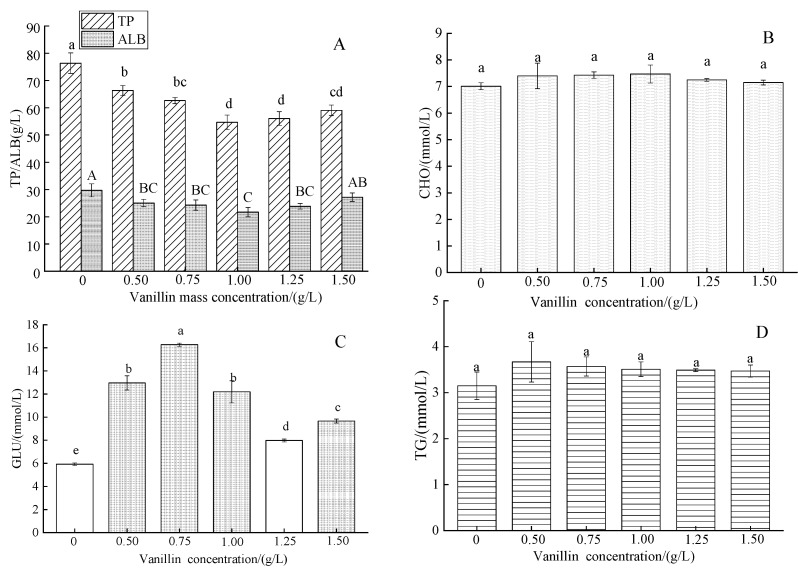
Effect of vanillin concentration on the blood serum concentrations of organic components in crucian carp: (**A**) total protein (TP) and albumin (ALB); (**B**) cholesterol (CHO); (**C**) glucose (GLU); and (**D**) triglycerides (TG). Different letters above the columns indicate a significant difference (*p* < 0.05); the same letter indicates no significant difference (*p* > 0.05).

**Figure 6 foods-12-01614-f006:**
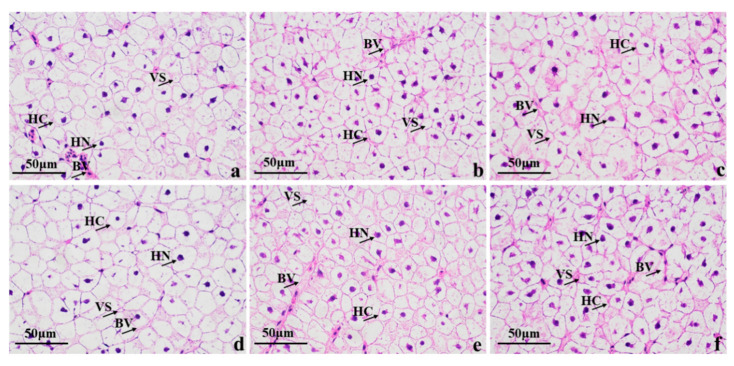
H&E-stained crucian carp liver tissue sections of (**a**) control group; (**b**–**f**) vanillin treatments at 0.50, 0.75, 1.00, 1.25, and 1.50 mg/L, respectively. Note: VS: vacuoles; HC: Hepatocytes; HN: Hepatocyte nucleus; BV: blood vessels.

**Figure 7 foods-12-01614-f007:**
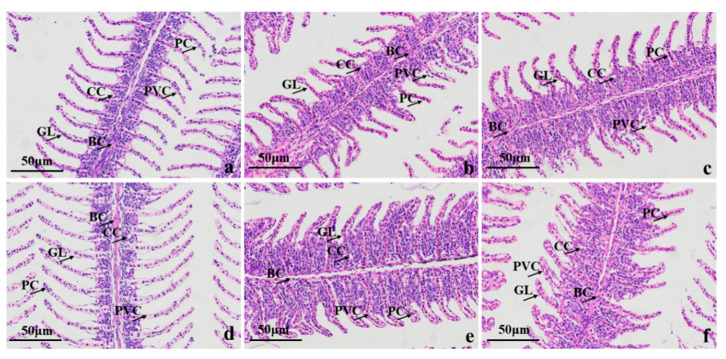
H&E-stained crucian carp gill tissue sections of (**a**) control group; (**b**–**f**) vanillin treatments at 0.50, 0.75, 1.00, 1.25, and 1.50 mg/L, respectively. Note: BC: blood corpuscle; CC: chlorine cells; GL: gill lamellae; PC: column cells; PVC: flat epithelial cells.

**Figure 8 foods-12-01614-f008:**
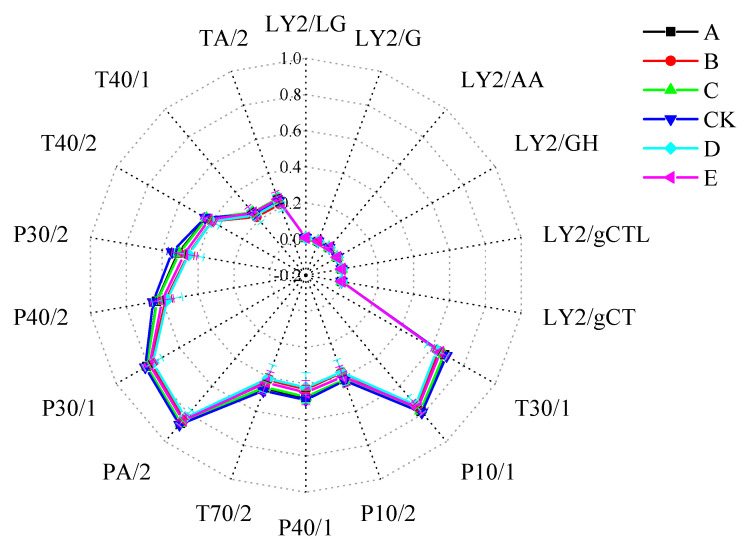
Effect of vanillin concentration on the radar fingerprint of E−Nose responses to carp-muscle flavour volatiles. Note: A, B, C, CK, D, and E represent vanillin concentrations of 0.50, 0.75, 1.00, 0 (control), 1.25, and 1.50 g/L, respectively.

**Figure 9 foods-12-01614-f009:**
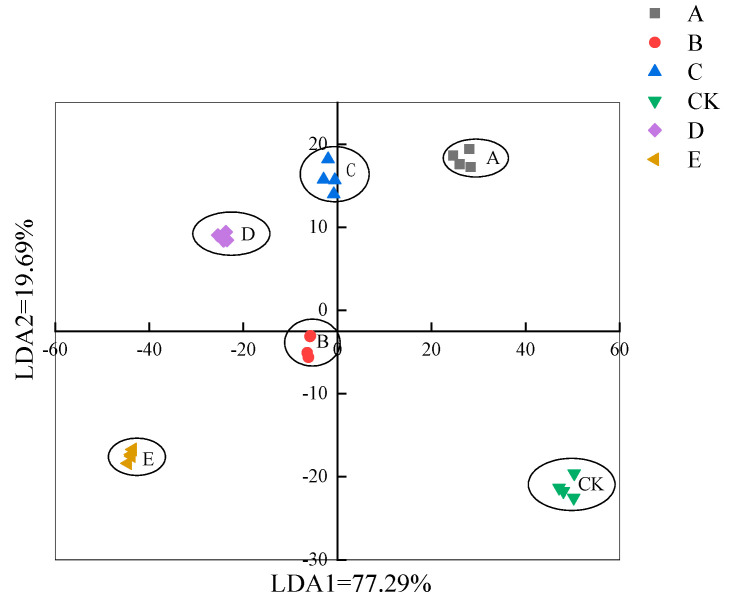
LDA of E−Nose response to crucian carp muscle at different vanillin treatment concentrations. Note: A, B, C, CK, D, and E represent vanillin concentrations of 0.50, 0.75, 1.00, 0 (control), 1.25, and 1.50 g/L, respectively.

**Figure 10 foods-12-01614-f010:**
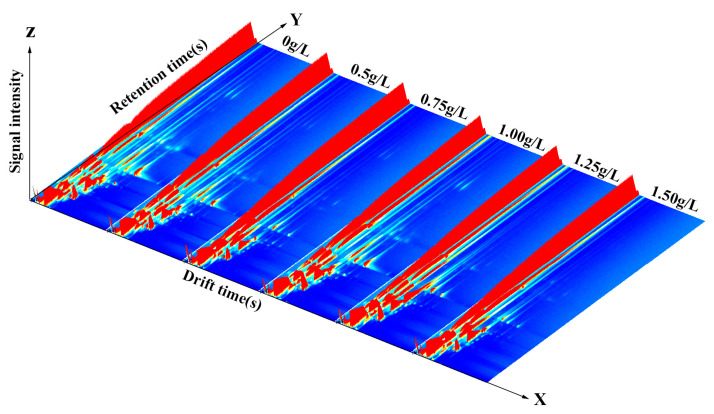
GC-IMS 3D spectra of crucian carp muscle after anaesthesia. From left to right: 0, 0.50, 0.75, 1.00, 1.25, and 1.50 g/L vanillin.

**Figure 11 foods-12-01614-f011:**
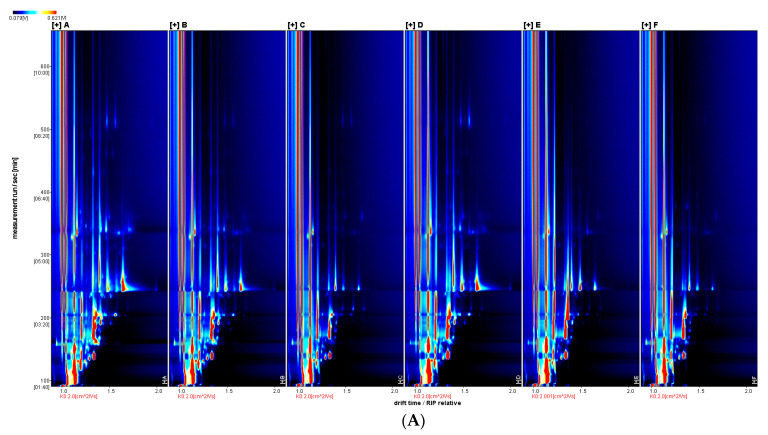
GC-IMS two-dimensional spectra of flavour volatiles resulting from different vanillin treatment concentrations on crucian carp muscle. Note: (**A**): top view; (**B**): difference plot. From left to right: 0, 0.50, 0.75, 1.00, 1.25, and 1.50 g/L vanillin.

**Figure 12 foods-12-01614-f012:**
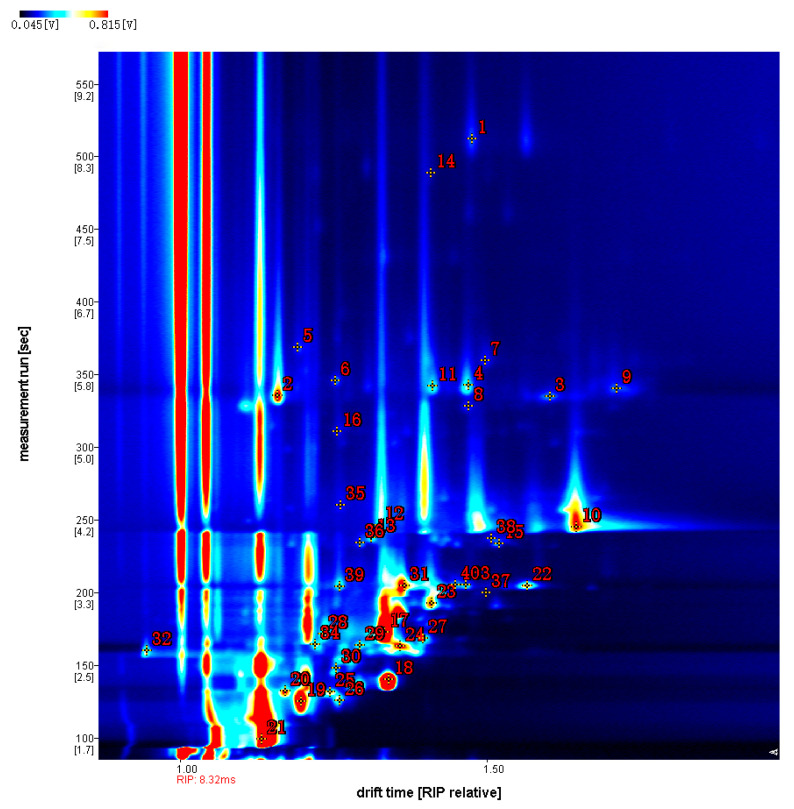
The 2D GC-IMS chromatogram of volatile organic compounds in crucian carp muscle, after treatment with vanillin. The numbers correspond to the identities in Table 4.

**Figure 13 foods-12-01614-f013:**
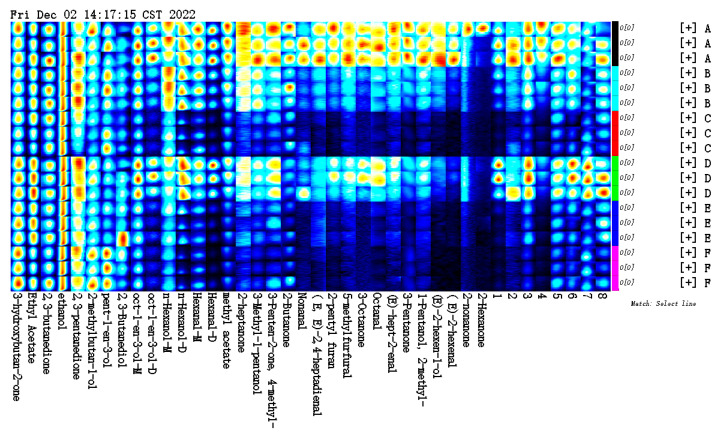
Fingerprint of volatile flavour compounds detected in crucian carp muscle after treatment with different concentrations of vanillin. Note: A-control; B-0.50 g/L; C-0.75 g/L; D-1.00 g/L; E-1.25 g/L; and F-1.50 g/L.

**Table 1 foods-12-01614-t001:** Response characteristics of E-Nose sensors.

Serial Number	Sensor Name	Sensor Response Characteristics
1	LY2/LG	chlorine, fluorine, nitrogen oxides, sulfides
2	LY2/G	ammonia, amine compounds, carbon oxides
3	LY2/AA	ethanol, acetone, ammonia
4	LY2/GH	ammonia, amine compounds
5	LY2/gCTL	hydrogen sulfide
6	LY2/gCT	propane, butane
7	T30/1	polar compounds, hydrogen chloride
8	P10/1	nonpolar; hydrocarbons, ammonia, chlorine
9	P10/2	nonpolar; methane, ethane
10	P40/1	fluorine, chlorine
11	T70/2	toluene, xylene, carbon monoxide
12	PA/2	ethanol, ammonia, amine compounds
13	P30/1	hydrocarbons, ammonia, ethanol
14	P40/2	chlorine, hydrogen sulfide, hydrogen fluoride
15	P30/2	hydrogen sulfide, ketones
16	T40/2	chlorine
17	T40/1	fluorine
18	TA/2	ethanol

**Table 2 foods-12-01614-t002:** Effect of different vanillin concentrations on the time taken to achieve anaesthesia and time taken for complete recovery for crucian carp.

Vanillin Concentration/(g/L)	Test the Number of Fish/Tail	Length/cm	Body Mass/g	Anaesthesia Time/min	Recovery Time/min	Recovery Rate/100%
0.25	10	16.1 ± 0.35	271.7 ± 37.86	33.00 ± 8.54	2.58 ± 0.13	100
0.50	10	16.3 ± 0.95	273.3 ± 79.11	2.48 ± 0.03	2.13 ± 0.12	100
0.75	10	16.6 ± 0.96	291.6 ± 68.07	1.64 ± 0.35	2.21 ± 0.22	100
1.00	10	15.8 ± 0.41	263.4 ± 70.71	1.42 ± 0.52	2.46 ± 0.07	100
1.25	10	16.6 ± 0.80	288.3 ± 25.65	1.36 ± 0.13	2.48 ± 0.02	100
1.50	10	17.8 ± 0.72	310.1 ± 13.22	1.24 ± 0.09	4.59 ± 0.27	100

**Table 3 foods-12-01614-t003:** Behavioural characteristics of crucian carp during the successive stages of anaesthesia and recovery.

Stages	Behavioural Characteristics	Minute
A_0_ stress period	A stress response occurs, swimming is accelerated, and operculum opening and closing are accelerated	0.40 ± 0.06
A_1_ sedation period	The response to external stimuli is weakened, the ability to swim is weakened, the body is slightly out of balance, and the breathing rate is further increased	0.75 ± 0.12
A_2_ Mild anaesthesia phase	The body rolls on its side, the ability to respond to external stimuli continues to weaken, swimming slowly, and the rate of operculum opening and closing decreases	1.09 ± 0.15
A_3_ anaesthesia period	The body of the fish is out of balance, ventral face up, stationary, and the operculum opening is reduced but continuous	2.60 ± 0.13
A_4_ deep anaesthesia period	The body of the fish is stationary, and the operculum opens and closes extremely slowly and irregularly	4.45 ± 0.42
R_1_ recovery stage 1	The ventral side of the fish is stationary and breathing begins to slowly resume continuously	1.44 ± 0.12
R_2_ recovery stage 2	The fish can swim slowly laterally, but the sense of direction is not clear, and the frequency of operculum opening is close to that before anaesthesia	1.91 ± 0.10
R_3_ recovery stage 3	The fish body was completely restored to its preanaesthesia state, and the operculum and upper and lower jaw opening frequency returned to normal	3.61 ± 0.18
R_4_ recovery stage 4	The operculum opens and closes normally, fully returns to normal swimming, and responds rapidly to stimuli	4.78 ± 0.27

**Table 4 foods-12-01614-t004:** Volatile organic compounds were identified from crucian carp muscle, after treatment with vanillin.

Category	Characteristic Peak Number	Name of the Compound	CAS#	Retention Index	Retention Time/min	Migration Time/min	Description of the Incense
Aldehydes	1	Nonanal	C124196	1107.9	512.244	1.47608	grease, cucumber and sweet orange flavours
	5	(E,E)-2,4-heptadienal	C4313035	1012.9	368.968	1.19232	aroma of grass and chicken
	8	5-methylfurfural	C620020	974.9	328.377	1.47055	cocoa, almonds
	11	Octanal	C124130	989.8	342.276	1.41258	fat, soap
	15	(E)-2-hexenal	C6728263	844.7	233.629	1.5202	fruity, green and vegetable
	16	(E)-hept-2-enal	C18829555	955.1	310.888	1.25698	aroma of grass and oil
	22	Hexanal-D	C66251	788.4	204.748	1.56613	grassy flavour
	39	Hexanal-M	C66251	787.5	204.354	1.26096	grassy flavour
Alcohols	2	1-octen-3-ol- M	C3391864	982.7	335.632	1.15976	mushroom, lavender, rose and hay aromas
	3	1-octen-3-ol-D	C3391864	982.2	335.107	1.60295	mushroom, lavender, rose and hay aromas
	10	*n*-Hexanol-D	C111273	865.2	245.133	1.64596	herbal flavour
	12	*n*-Hexanol-M	C111273	867	246.147	1.32588	herbal flavour
	13	3-Methyl-1-pentanol	C589355	852.3	237.822	1.31307	fermented taste
	21	ethanol	C64175	421.1	99.501	1.13351	alcohol
	28	2-methylbutan-1-ol	C137326	714.7	171.67	1.23403	aromatic with wine and ether
	31	2,3-Butanediol	C513859	788.4	204.748	1.36608	fermented taste
	32	pent-1-en-3-ol	C616251	684.2	159.959	0.94764	fruity aroma
	36	1-Pentanol,2-methyl	C105306	845.8	234.228	1.29357	fermented taste
	38	(E)-2-hexen-1-ol	C928950	851.7	237.511	1.50757	grassy, fruity
Ketones	9	3-Octanone	C106683	987.9	340.428	1.71197	fruity aroma
	14	2-nonanone	C821556	1094.4	488.937	1.40882	fruity, sweet and green notes
	17	3-hydroxybutan-2-one	C513860	717.3	172.744	1.33455	aromatic smell
	20	2,3-butanedione	C431038	580.2	132.579	1.17293	Fermented aroma, sweet aroma
	24	3-Pentanone	C96220	693.3	163.079	1.36017	sweet scent
	25	2-Butanone	C78933	576.6	131.72	1.24487	aromatic smell
	34	2,3-pentanedione	C600146	697	164.522	1.22074	Caramel aroma, diluted with a creamy smell
	35	2-heptanone	C110430	890.4	260.017	1.26204	fruity aroma
	37	2-Hexanone	C591786	779	200.268	1.50009	spicy smell
	40	3-Penten-2-one, 4-methyl	C141797	790.2	205.63	1.45009	sweet scent
Esters	18	ethyl acetate	C141786	610.7	140.096	1.34145	pineapple flavour
	19	methyl acetate	C79209	548.8	125.276	1.19757	pineapple flavour
Furans	6	2-pentyl furan	C3777693	993.6	345.924	1.25402	fruity, grassy

Note: M indicates monomer and D indicates dimer.

## Data Availability

The data are available from the corresponding author.

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
