# Peer review of "Effect of Vanillin on the Anaesthesia of Crucian Carp: Effects on Physiological and Biochemical Indices, Pathology, and Volatile Aroma Components"

_foods, 2023, doi:10.3390/foods12081614_

Round 1
Reviewer 2 Report
The authors have addressed the effects of vanillin on anaesthesia of crucian carp through physiological and biochemical indices, its pathology, and muscle aroma after treatment.
The paper is of interest to the general public, it is well written and well conceived, the experiments are adequate. The only shortcoming is the small number of specimens used for biochemical blood analyses.
Minor comments:
Ln 40 Freshwater
Ln. 46 Tricaine methane-sulphonate (MS-222), C9H11O2N + CH3SO3H, is also known as ethyl m-amino benzoate, tricaine mesilate, m-aminobenzoic acid ethyl ester methanesulfonate and metacaine, so you can use any of this names to describe it. 3-aminobenzoic acid ethyl ester methanesulfonate is not a proper denomination.
Ln 60 Vanillin is commonly known also as methyl vanillin or vanillic aldehyde, but has also dozens of other names (https://pubchem.ncbi.nlm.nih.gov/compound/Vanillin#section=Environmental-Fate-Exposure-Summary). Vanillin has been reported in the essential oil of Java citronella, but citronella is not its synonym. Therefore, delete “also known as citronella”
Ln 91 insert latin name for crucian carp
Ln 111 Ten carps were used for each treatment? How many animals in total were used? Please insert.
Ln 130 Three fish for blood biochemistry parameters is extremely low. Please comment the rationale for using only three fish
Ln 147 What was the thickness of the microtomed sections? Please insert.
Overall, would you recommend the vanillin for fish anaesthesia regarding the muscle flavor of the treated fish?
